# Cyclin D activates the Rb tumor suppressor by mono-phosphorylation

**Anil M Narasimha[1†], Manuel Kaulich[1†], Gary S Shapiro[1†‡], Yoon J Choi[2,3], Piotr Sicinski[2,3], Steven F Dowdy[1]\***

[1]Department of Cellular and Molecular Medicine, University of California, San Diego School of Medicine, La Jolla, United States; [2]Department of Genetics, Harvard Medical School, Boston, United States; [3]Department of Cancer Biology, Dana-Farber Cancer Institute, Boston, United States

**Abstract** The widely accepted model of $G_1$ cell cycle progression proposes that cyclin D:Cdk4/6 inactivates the Rb tumor suppressor during early $G_1$ phase by progressive multi-phosphorylation, termed hypo-phosphorylation, to release E2F transcription factors. However, this model remains unproven biochemically and the biologically active form(s) of Rb remains unknown. In this study, we find that Rb is exclusively mono-phosphorylated in early $G_1$ phase by cyclin D:Cdk4/6. Mono-phosphorylated Rb is composed of 14 independent isoforms that are all targeted by the E1a oncoprotein, but show preferential E2F binding patterns. At the late $G_1$ Restriction Point, cyclin E:Cdk2 inactivates Rb by quantum hyper-phosphorylation. Cells undergoing a DNA damage response activate cyclin D:Cdk4/6 to generate mono-phosphorylated Rb that regulates global transcription, whereas cells undergoing differentiation utilize un-phosphorylated Rb. These observations fundamentally change our understanding of $G_1$ cell cycle progression and show that mono-phosphorylated Rb, generated by cyclin D:Cdk4/6, is the only Rb isoform in early $G_1$ phase.

**\*For correspondence:** sdowdy@ucsd.edu

[†]These authors contributed equally to this work

**Present address:** [‡]Sanofi Oncology, Cambridge, United States

**Competing interests:** The authors declare that no competing interests exist.

**Reviewing editor**: Roger Davis, University of Massachusetts Medical School, United States

## Introduction

The retinoblastoma tumor suppressor protein (Rb) functions to regulate multiple critical cellular activities, including the late $G_1$ restriction point, the DNA damage response checkpoint, cell cycle exit, and differentiation (*Burkhart and Sage, 2008*; *Paternot et al., 2010*; *Henley and Dick, 2012*; *Johnson and Skotheim, 2013*). However, the Rb gene is infrequently mutated or deleted, instead upstream pathways that regulate Rb by phosphorylation on Cdk sites are altered in the majority of human cancers, including deletion and mutation of the p16 tumor suppressor and upregulation and mutation of cyclin D1, D2, D3, Cdk4 and Cdk6 genes (*Sherr and McCormick, 2002*; *Knudsen and Knudsen, 2006*; *Burkhart and Sage, 2008*; *Henley and Dick, 2012*; *Choi and Anders, 2013*). Rb contains 16 putative Cdk phosphorylation sites that are spread throughout the protein, and all but one (S567) lie outside of Rb's structured A'/B' and A/B-box or 'pocket' protein–protein binding domains (Figure 1A). Rb is thought to exist in three generalized isoforms: (1) un-phosphorylated Rb; (2) hypo-phosphorylated Rb, also referred to as 'under' phosphorylated Rb or 'partially' phosphorylated Rb; and (3) inactive hyper-phosphorylated Rb, present in late $G_1$, S, $G_2$ and M phases that is readily identifiable by SDS-PAGE as a slower migrating species (*Burkhart and Sage, 2008*; *Paternot et al., 2010*; *Henley and Dick, 2012*). Surprisingly, given the scientific scrutiny of Rb over the last 25 years, the biochemical identification of the biologically active isoform(s) of Rb required for early $G_1$ phase regulation, DNA damage checkpoint control, cell cycle exit, and differentiation remains unknown.

To dissect Rb function and regulation, many early reports utilized supra-physiologic overexpression studies using various cyclins (A, B, D, E) and Cdks (−1, −2, −4, −6) that resulted in Rb inactivation by hyper-phosphorylation associated with an accelerated S-phase entry, and induction of E2F-dependent

**eLife digest** Cells go through a tightly controlled, multi-step procedure before they divide. This cell division program—the cell cycle—is necessary for preventing unrestrained cellular growth, which may lead to cancer. Proteins called cyclins control the progression through each of the phases of the cell cycle, with different cyclins working during different phases.

During the G1 phase of the cell cycle, cells grow in size and produce the proteins that are required to copy DNA. Once a cell passes a checkpoint called the 'restriction point' at the end of the G1 phase, it is committed to dividing. It is therefore particularly important to keep events during G1 phase in check.

The Retinoblastoma tumor suppresor protein (Rb) is a key player in regulating the G1 phase. Rb sequesters transcription factors that are essential for the cell cycle to progress. Previously, it was thought that a complex called cyclin D added more and more phosphates to the Rb protein during the G1 phase. This process predicted a slow release of transcription factors, which attach to DNA and start the process of DNA replication. While many studies have presented data that is consistent with this model, direct biochemical evidence of these events is lacking.

Narasimha, Kaulich, Shapiro et al. now present biochemical analyses of Rb proteins that show—completely unexpectedly—that the cyclin D complex adds just one phosphate group to Rb during the G1 phase, although this group can be added to one of fourteen different sites. The resulting 'mono-phosphorylated' Rb varieties can each sequester different transcription factors and stop them working.

At the restriction point, many more phosphate groups are then rapidly added, and the Rb protein is inactivated by a different cyclin. This cyclin—called Cyclin E—then drives cells into the next phase of the cell cycle. Establishing how cyclin E is activated is a priority for future research.

target genes (*Hinds et al., 1992*; *Ewen et al., 1993*; *Resnitzky et al., 1994*; *Lundberg and Weinberg, 1998*). Likewise, supra-physiologic overexpression studies using Rb constructs where many, but not all, of the putative Cdk Ser/Thr consensus sites were mutated to Ala residues resulted in repressed E2F-dependent transcription and cell cycle arrest, as did overexpression of Cdk inhibitors, p16, p21, and p27 (*Sherr, 1994*; *Knudsen and Wang, 1997*; *Leng et al., 1997*; *Sherr and McCormick, 2002*; *Knudsen and Knudsen, 2006*; *Burkhart and Sage, 2008*; *Paternot et al., 2010*; *Henley and Dick, 2012*; *Choi and Anders, 2013*). Collectively, over the last 20 years, these studies have led to a widely accepted model of $G_1$ cell cycle progression that proposes cyclin D:Cdk4/6 inactivates Rb during early $G_1$ phase by progressive multi-phosphorylation, termed 'hypo-phosphorylation', resulting in release of E2F transcription factors that induce expression of cyclin E, resulting in activation of cyclin E:Cdk2 complexes that complete Rb inactivation by hyper-phosphorylation in late $G_1$ phase. The key tenet of this model is the progressive multi-phosphorylating, hypo-phosphorylation of Rb by cyclin D:Cdk4/6 complexes; however, the putative hypo-phosphorylated Rb and un-phosphorylated Rb co-migrate on 1D SDS-PAGE and cannot be separated (*Ezhevsky et al., 2001*). Moreover, there is no biochemical data defining the extent or timing of phosphorylation that constitutes hypo-phosphorylated Rb. Consequently, it remains entirely unknown if hypo-phosphorylated Rb contains one, two, three, five, seven or more phosphates and at what phosphate number does the putative hypo-phosphorylated Rb become inactive to release E2F transcription factors. Thus, the critical core tenet of the $G_1$ model that cyclin D:Cdk4/6 inactivates Rb by progressive multi-phosphorylation or hypo-phosphorylation remains unproven biochemically.

We have previously found in kinetic analyses from highly synchronized normal cells and p16-deficient cancer cells that cyclin D:Cdk4/6 is constitutively active throughout early $G_1$ phase at the same time when Rb is repressing E2F target genes (*Ezhevsky et al., 1997*, *2001*; *Haberichter et al., 2007*). In fact, we only observed induction of E2F target genes upon the activation of cyclin E:Cdk2 complexes and the appearance of hyper-phosphorylated Rb. These observations questioned the core biological consequences of cyclin D:Cdk4/6 progressive hypo-phosphorylation of Rb during early $G_1$ phase. Here, for the first time, we separated all Rb isoforms by two-dimensional isoelectric focusing (2D IEF) and find that Rb is exclusively mono-phosphorylated in early $G_1$ phase in both normal and p16-deficient tumor cells. We found no experimental evidence to support the notion of progressive

multi-phosphorylating hypo-phosphorylation of Rb. Using Cdk4/6-specific inhibitors and triple cyclin D-deleted MEFs, we determined that cyclin D:Cdk4/6 is the Rb mono-phosphorylating kinase that generates 14 independent mono-phosphorylated Rb isoforms in early $G_1$ phase. At the late $G_1$ Restriction Point, activation of cyclin E:Cdk2 complexes perform a quantum hyper-phosphorylating inactivation of all mono-phosphorylated Rb isoforms. Cells undergoing a DNA damage response activate cyclin D:Cdk4/6 complexes to generate active mono-phosphorylated Rb that regulates global transcription, whereas cells exiting the cell cycle use un-phosphorylated Rb. Together, our observations demonstrate that mono-phosphorylated Rb, generated by cyclin D:Cdk4/6 complexes, is the functionally active Rb isoform present in early $G_1$ phase.

## Results

### Un-phosphorylated, mono-phosphorylated, and hyper-phosphorylated Rb isoforms

Rb contains 15 putative Cdk phosphorylation sites located on loops between or after structured A'/B' and A/B pocket domains (*Burke et al., 2012*; *Lamber et al., 2013*) (*Figure 1A*). Rb is thought to exist in three generalized biochemical states: un-phosphorylated Rb; progressive hypo-phosphorylated Rb (also termed 'under' or 'partially' phosphorylated Rb); and inactive hyper-phosphorylated Rb (*Lee et al., 1987*; *DeCaprio et al., 1989*; *Ludlow et al., 1989*; *Mittnacht et al., 1994*; *Ezhevsky et al., 2001*). Although early $G_1$ phase Rb hypo-phosphorylation was first reported 25 years ago (*Ludlow et al., 1989*), the actual number, kinetics and location of phosphates on hypo-phosphorylated Rb remains entirely unknown. Analysis of synchronized primary human foreskin fibroblasts (HFFs) arrested in early $G_1$ phase by contact inhibition in the presence of serum and released by replating at low density progressed through early $G_1$ phase with constitutively active cyclin D:Cdk4/6 complexes and no evidence for increased Rb phosphorylation or transcriptional induction of Cdc6, a key E2F target gene repressed by active Rb (*Morris and Dyson, 2001*) (*Figure 1B*). In contrast, Cdc6 was induced ten-fold upon cyclin E:Cdk2 activation and Rb hyper-phosphorylation, which migrates more slowly on 1D SDS-PAGE. However, the putative hypo-phosphorylated Rb isoforms co-migrated as a single fastest migrating species during all of the early $G_1$ phase time points (*Figure 1B*). Thus, there is either no evidence for progressive hypo-phosphorylation of Rb and/or 1D SDS-PAGE is not capable of separating all Rb phospho-isoforms.

Phosphates are highly acidic modifications that significantly change the isoelectric point (pI) of a protein. Unlike 1D SDS-PAGE, two-dimensional isoelectric focusing (2D IEF) can separate specific phospho-isoforms of a protein based on total phosphate numbers regardless of position within the protein or nature of the modified residue (*Figure 1C*). Therefore, we utilized 2D IEF to ascertain the extent and kinetics of the putative progressive multi-phosphorylated hypo-phosphorylation of Rb during early $G_1$ phase. First, we calibrated the 2D IEF by generating a non-phosphorylatable Rb construct (ΔCdk Rb) standard where 15 of the 16 putative Cdk sites were converted to Ala residues (*Figure 1A*), plus we added an N-terminal HA tag. We left S567 unaltered because it is buried in the central core of Rb's A-box and solvent inaccessible (*Lee et al., 1998*). The isoelectric point of un-phosphorylated Rb is 8.1, and 2D IEF of the ΔCdk Rb construct expressed in asynchronous cycling cells focused as a single basic species with a pI ~8 (*Figure 1D*), confirming that Rb is only phosphorylated on Cdk sites and that S567 is not phosphorylated in vivo.

Starting with ΔCdk Rb, we generated Rb phospho-isoform standards by restoring one (Rb[1xCdk]), two (Rb[2xCdk]), three (Rb[3xCdk]), six (Rb[6xCdk]), nine (Rb[9xCdk]), or fifteen (Rb[15xCdk]) Cdk sites on Rb. 2D IEF of single Cdk site Rb[1xCdk] construct expressed in cycling cells focused as a single phosphorylated Rb species with a more acidic pI ~7.0, that we termed mono-phosphorylated Rb (*Figure 1D*). The double Cdk site Rb[2xCdk] construct focused as two spots: mono-phosphorylated Rb and di-phosphorylated Rb. Surprisingly, 2D IEF of the Rb[3xCdk] construct focused as mono-phosphorylated and tri-phosphorylated Rb, with no di-phosphorylated Rb (*Figure 1D*). Likewise, 2D IEF of Rb constructs containing six (Rb[6xCdk]), nine (Rb[9xCdk]) and fifteen (Rb[15xCdk]) Cdk sites resulted in the appearance of mono-phosphorylated Rb, plus either a six, nine, or >12 phosphate (pI < 4) Rb species, respectively, and the absence of any intermediate Rb phospho-isoforms (*Figure 1D*). Thus, unlike 1D SDS-PAGE, 2D IEF quantitatively separated all Rb isoforms from un-phosphorylated Rb to mono-phosphorylated Rb and all multi-phosphorylated Rb isoforms up to hyper-phosphorylated Rb.

We next analyzed endogenous, wild-type Rb from primary HFFs by 2D IEF. Consistent with no [32]P-labeling of Rb from $G_0$ arrested cells (*DeCaprio et al., 1989*; *Ezhevsky et al., 2001*), Rb from

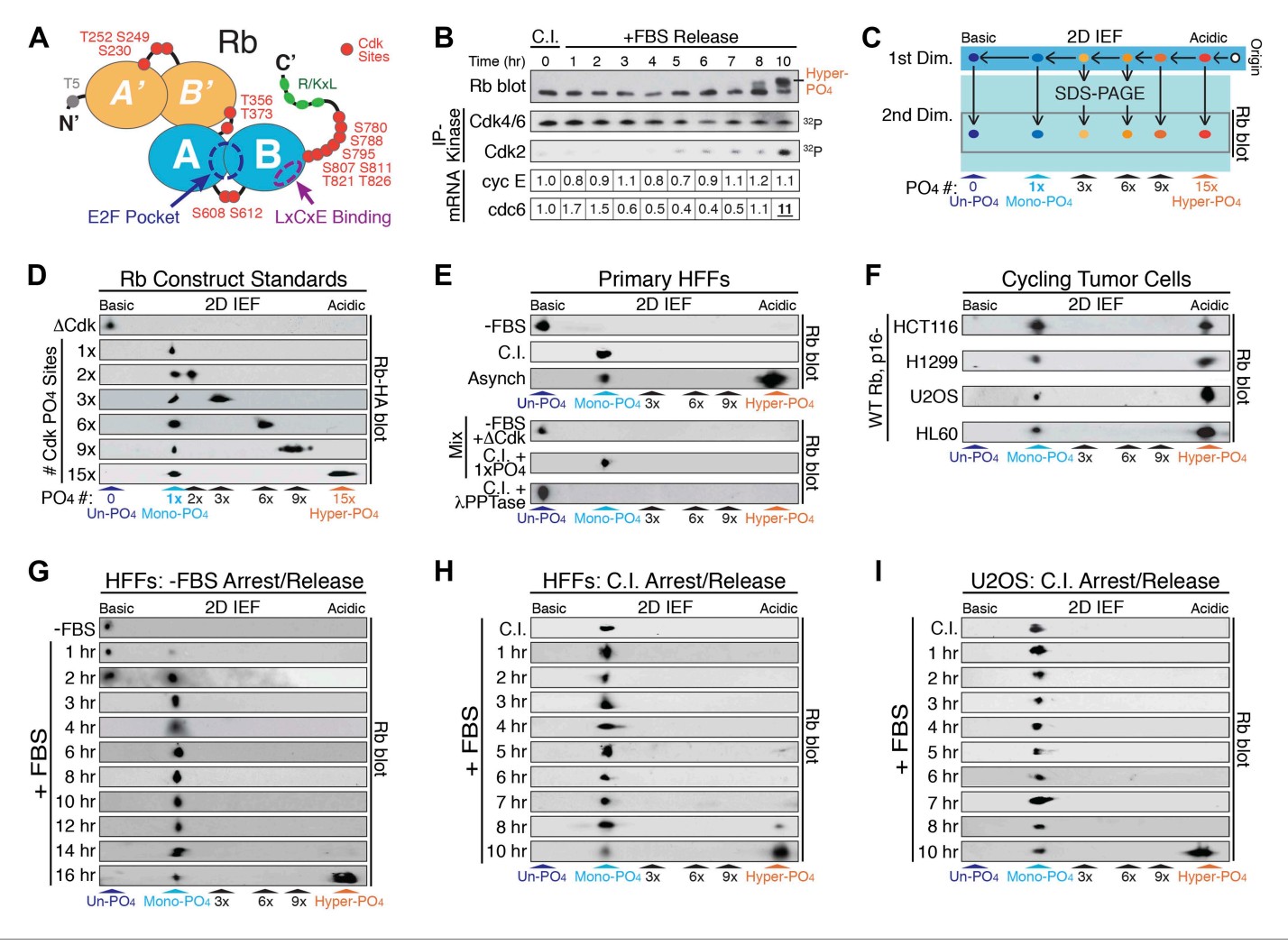

**Figure 1**. Rb is exclusively mono-phosphorylated in early $G_1$ phase. (**A**) Schematic diagram of human Rb Cdk phosphorylation sites, A'/B' and A/B pocket domains. (**B**) Kinetic analysis of contact inhibited early $G_1$ phase arrested (+FBS) and released primary Human Fibroblasts (HFFs) by 1D SDS-PAGE Rb immunoblot, anti-Cdk4/6, and anti-Cdk2 immunoprecipitation-kinase assay, and qRT-PCR of cyclin E and cdc6 mRNA normalized to B-2-microglobulin levels. (**C**) Schematic diagram of two-dimensional isoelectric focusing (2D IEF). Immunoprecipitated Rb is loaded at origin on acidic end of IEF strip and separated first by pI. IEF strip is then soaked in SDS, run in second dimension into SDS-PAGE and immunoblotted for Rb. (**D**) 2D IEF Rb-HA immunoblot of Rb construct standards expressed in cycling cells and containing 0 (ΔCdk), 1x, 2x, 3x, 6x, 9x or 15x Cdk phosphorylation sites. (**E**) Top panels: 2D IEF Rb immunoblot of primary HFFs serum deprived $G_0$ arrested (−FBS), contact inhibited early $G_1$ phase arrested (+FBS), or asynchronously cycling. Bottom panels: 2D IEF Rb immunoblot of serum deprived $G_0$ arrested (−FBS) HFFs mixed with ΔCdk Rb standard, contact inhibited early $G_1$ phase arrested (+FBS) HFFs mixed with single Cdk site Rb standard and contact inhibited treated with λ phosphatase. (**F**) 2D IEF Rb immunoblot from cycling human tumor cell lines expressing wild-type Rb and deregulated cyclin D:Cdk4/6 due to p16 deletion, HCT116 colon carcinoma, H1299 lung adenocarcinoma, U2OS osteosarcoma, HL60 promyelocytic leukemia. (**G**) 2D IEF Rb immunoblot from serum deprived $G_0$ arrested (−FBS) and released (+FBS) primary HFFs from 0 to 16 hr (**H**) 2D IEF Rb immunoblot from contact inhibited early $G_1$ phase arrested (+FBS) and released HFFs from 0 to 10 hr (**I**) 2D IEF Rb immunoblot from contact inhibited early $G_1$ phase arrested (+FBS) and released U2OS from 0 to 10 hr.

The following figure supplements are available for figure 1:

**Figure supplement 1**. Rb is exclusively mono-phosphorylated in early $G_1$ phase.

serum deprived $G_0$ arrested primary HFFs focused as a single, basic isoform with a pI ~8 (*Figure 1E*). 2D IEF of Rb from $G_0$-arrested primary HFFs mixed with non-phosphorylatable ΔCdk Rb standard, co-focused as a single un-phosphorylated Rb species. In contrast, Rb from contact inhibited early $G_1$ phase arrested (+FBS) HFFs focused as a single mono-phosphorylated species with a pI ~7.0 (*Figure 1E*). 2D IEF of Rb from contact inhibited HFF Rb mixed with the single $Rb^{1xCdk}$ construct standard confirmed

that contact inhibited cells contain only mono-phosphorylated Rb. Lambda phosphatase treatment of contact inhibited mono-phosphorylated Rb collapsed into un-phosphorylated Rb (*Figure 1E*). 2D IEF of Rb from asynchronously cycling primary HFFs focused as two isoforms: mono-phosphorylated Rb plus a very acidic hyper-phosphorylated Rb isoform (pI < 4) (*Figure 1E*).

The majority of human tumors expressing wild-type Rb contain oncogenic mutations that upregulate cyclin D:Cdk4/6 kinase activity (*Sherr and McCormick, 2002*; *Choi and Anders, 2013*). We examined four disparate human tumor cell lines that express wild-type Rb, and are deleted for the p16 tumor suppressor gene, a specific inhibitor of Cdk4/6. Surprisingly, 2D IEF of Rb from cycling populations of all four tumor cell lines, HCT116 colon carcinoma, H1299 lung adenocarcinoma, U2OS osteosarcoma, and HL60 promyelocytic leukemia, showed the presence of only mono-phosphorylated Rb and hyper-phosphorylated Rb, with no evidence of multi-phosphorylated, hypo-phosphorylated Rb isoforms, even though cyclin D:Cdk4/6 was deregulated in all four tumor cell types (*Figure 1F*). Together, these results present several significant insights into the biochemical properties of Rb phosphorylation in vivo. First, $G_0$-arrested cells contain un-phosphorylated Rb. Second, Rb from both cycling normal and p16-deleted tumor cells is only mono-phosphorylated or hyper-phosphorylated in vivo. Third, these observations point to phosphorylation of Rb by two entirely independent cyclin:Cdk activities: (1) a Rb mono-phosphorylating Cdk activity that places one, and only one, phosphate on Rb; and (2) a Rb hyper-phosphorylating Cdk activity that places >12 phosphates on Rb.

## Rb is exclusively mono-phosphorylated in early $G_1$ phase of normal and tumor cells

The current widely accepted model of $G_1$ cell cycle progression proposes that Rb becomes progressively more hypo-phosphorylated by cyclin D:Cdk4/6 complexes as cells advance through early $G_1$ phase. To test this notion, we performed kinetic analyses on HFF cells arrested in $G_0$ by serum deprivation and restimulated by serum addition (+FBS) to enter early $G_1$ phase (*Figure 1G*). $G_0$ cells contained only un-phosphorylated Rb, but by 1 hr post-stimulation, a small amount of mono-phosphorylated was detected, concurrent with activation of cyclin D:Cdk4/6 complexes (*Figure 1—figure supplement 1A*). By 3 hr, only mono-phosphorylated Rb was present. Surprisingly, Rb remained exclusively mono-phosphorylated throughout the entire early $G_1$ phase time points at 3, 4, 5, 6, 8, 10, 12. and 14 hr with no higher order phosphorylation species detected (*Figure 1G*), even though cyclin D:Cdk4/6 complexes were constitutively active. At 16 hr post-release, we detected a quantum switch-like shift to hyper-phosphorylated Rb concomitant with activation of cyclin E:Cdk2 complexes (*Figure 1—figure supplement 1A*). We next performed a kinetic analysis on primary HFFs arrested in early $G_1$ phase by contact inhibition (+FBS) and released by replating at low density. Rb remained exclusively mono-phosphorylated throughout the entire early $G_1$ phase time points from 0 to 7 hr with no higher order Rb phosphorylation species detected (*Figure 1H*). Cyclin D:Cdk4/6 kinase activity was constitutively active in contact arrested early $G_1$ phase cells and throughout all of early $G_1$ phase with no detectable cyclin E:Cdk2 kinase activity (*Figure 1B*). At 10 hr post-release, we detected a strong shift to hyper-phosphorylated Rb (>12 phosphates), concomitant with activation of cyclin E:Cdk2 complexes and transcriptional induction of cdc6, an E2F target gene (*Figure 1B*). Again, we surprisingly did not detect any evidence for the progressive multi-phosphorylation or hypo-phosphorylation of Rb in early $G_1$ phase or increases in E2F target genes, even though cyclin D:Cdk4/6 complexes were constitutively active.

We next examined synchronized p16-deleted human U2OS osteosarcoma tumor cells (*Figure 1I*). Contact inhibited early $G_1$-arrested U2OS cells (+FBS) contained only mono-phosphorylated Rb, with no higher order Rb phosphorylated species, and active cyclin D:Cdk4/6 complexes (*Figure 1—figure supplement 1B*). Rb remained exclusively mono-phosphorylated throughout all of the early $G_1$ phase time points at 1, 2, 3, 4, 5, 6, 7, and 8 hr in the presence of constitutively active cyclin D:Cdk4/6 complexes with no evidence of progressive hypo-phosphorylation. We first detected hyper-phosphorylated Rb at 10 hr post-release (*Figure 1I*). Together, these observations demonstrated that both primary and tumor cells exclusively generate mono-phosphorylated Rb during all of early $G_1$ phase before being converted in a quantum step to hyper-phosphorylated Rb at the late $G_1$ phase Restriction Point. Collectively, we performed hundreds of 2D IEFs on Rb from 11 cell types and found no biochemical evidence to support the notion of progressive multi-phosphorylation or hypo-phosphorylation of Rb in early $G_1$ phase.

# Early G$_1$ phase cells contain fourteen independent mono-phosphorylated Rb isoforms

In all of our 2D IEFs of Rb, we detected one, and only one, phosphate on Rb during early G$_1$ phase. However, *Mittnacht et al. (1994)* reported that tryptic phospho-peptide mapping (where the $^{32}$P-labeled protein is cleaved into small peptides by trypsin digestion and then separated by charge and hydrophobicity) of total Rb isolated from early G$_1$ phase cells (labeled as hypo-phosphorylated Rb) retained the vast majority of the same phospho-peptide spots that hyper-phosphorylated Rb contained. In light of our new observations showing only mono-phosphorylated Rb present in early G$_1$ phase, the *Mittnacht et al. (1994)* study suggested the potential for the presence of many mono-phosphorylated Rb isoforms that when summed together would result in the observed phospho-peptide pattern.

To ascertain how many of Rb's 15 Cdk sites (*Figure 1A*) are mono-phosphorylated by cyclin D:Cdk4/6 in early G$_1$ phase, we used a series of phospho-specific Rb antibodies to immunoblot Rb from HFF cells arrested in G$_0$ by serum deprivation (un-phosphorylated Rb) and early G$_1$ phase arrested by contact inhibition (mono-phosphorylated Rb) (*Figure 2A*). While none of the antibodies recognized un-phosphorylated Rb from G$_0$ arrested HFFs, all of the phospho-specific Rb antibodies recognized mono-phosphorylated Rb isoforms from early G$_1$ phase arrested cells, including S249/S252, T373, S608, S612, S795, S807/S811, T821 and T826 (*Figure 2A*). Similar mono-phosphorylated Rb isoform results were obtained from contact inhibited early G$_1$ phase arrested p16-deficient U2OS cells (*Figure 2A*). We also note that all of the phospho-specific antibodies recognized hyper-phosphorylated Rb from S phase arrested cells (*Figure 2—figure supplement 1*). Together, these observations suggested the presence of at least 8 individual mono-phosphorylated Rb isoforms.

To independently confirm the presence of mono-phosphorylated Rb isoforms, we immunoprecipitated mono-phosphorylated Rb from contact inhibited early G$_1$ phase arrested HFFs with either the T826 or S608 phospho-specific Rb antibodies, and then immunoblotted with five phospho-specific Rb antibodies (*Figure 2B*). Consistent with the 2D IEF data, the phospho-specific immunoprecipitation of T826 mono-phosphorylated Rb was only recognized by immunoblot with the T826 phospho-specific antibody and not by T373, S608, S612, or S795 phospho-specific antibodies. Likewise, immunoprecipitation of S608 mono-phosphorylated Rb was only recognized by the S608 phospho-specific antibody and not by the other phospho-specific antibodies. In contrast, immunoprecipitated hyper-phosphorylated Rb from S phase arrested HFF cells with either the T826 or S608 phospho-specific antibodies was recognized by multiple phospho-specific antibodies (*Figure 2B*), supporting the presence of multiple phosphates on individual hyper-phosphorylated Rb molecules. To identify the extent of mono-phosphorylated Rb isoforms, we generated all 15 individual single Cdk site Rb constructs. 2D IEFs on each of the single Cdk site Rb constructs expressed in cells determined that 14 of the single Cdk site Rb constructs were mono-phosphorylated in vivo (*Figure 2C*). T5, which is not evolutionary conserved below primates, was not phosphorylated. Together, these observations demonstrate the presence of 14 independent mono-phosphorylated Rb isoforms in early G$_1$ phase and explain the large number of Rb tryptic phospho-peptide spots observed by *Mittnacht et al. (1994)*. Moreover, because early G$_1$ phase cells exclusively contain mono-phosphorylated Rb, by definition, some, most or all of the 14 mono-phosphorylated Rb isoforms must be biologically active.

Rb has been shown to bind to four members of the E2F family of transcription factors (E2F1-4) and over 100 additional cellular proteins (*Morris and Dyson, 2001*). We hypothesized that the generation of 14 mono-phosphorylated Rb isoforms may serve as a post-translational mechanism to diversify Rb from a single un-phosphorylated protein in G$_0$ into 14 independently functionalized mono-phosphorylated Rb isoforms that each bind specific cellular targets during early G$_1$ phase. To test this hypothesis, we independently co-transfected each single Cdk site mono-phosphorylated Rb-HA construct and control un-phosphorylatable ΔCdk Rb-HA into cells co-expressing the E1a oncoprotein or E2F-1, E2F-2, E2F-3, E2F-4 transcription factors (Myc tagged), then individually immunoprecipitated E1a and each E2F, and immunoblotted for the associated Rb mono-phosphorylated isoforms (*Figure 2D*). Given it's role in driving adenovirus infected quiescent G$_0$ cells (containing un-phosphorylated Rb) into G$_1$ phase and then S phase, it was not too surprising that the E1a oncoprotein bound equally well to un-phosphorylated Rb and all 14 mono-phosphorylated Rb isoforms. This observation also showed that all single Cdk site Rb constructs were correctly folded in vivo.

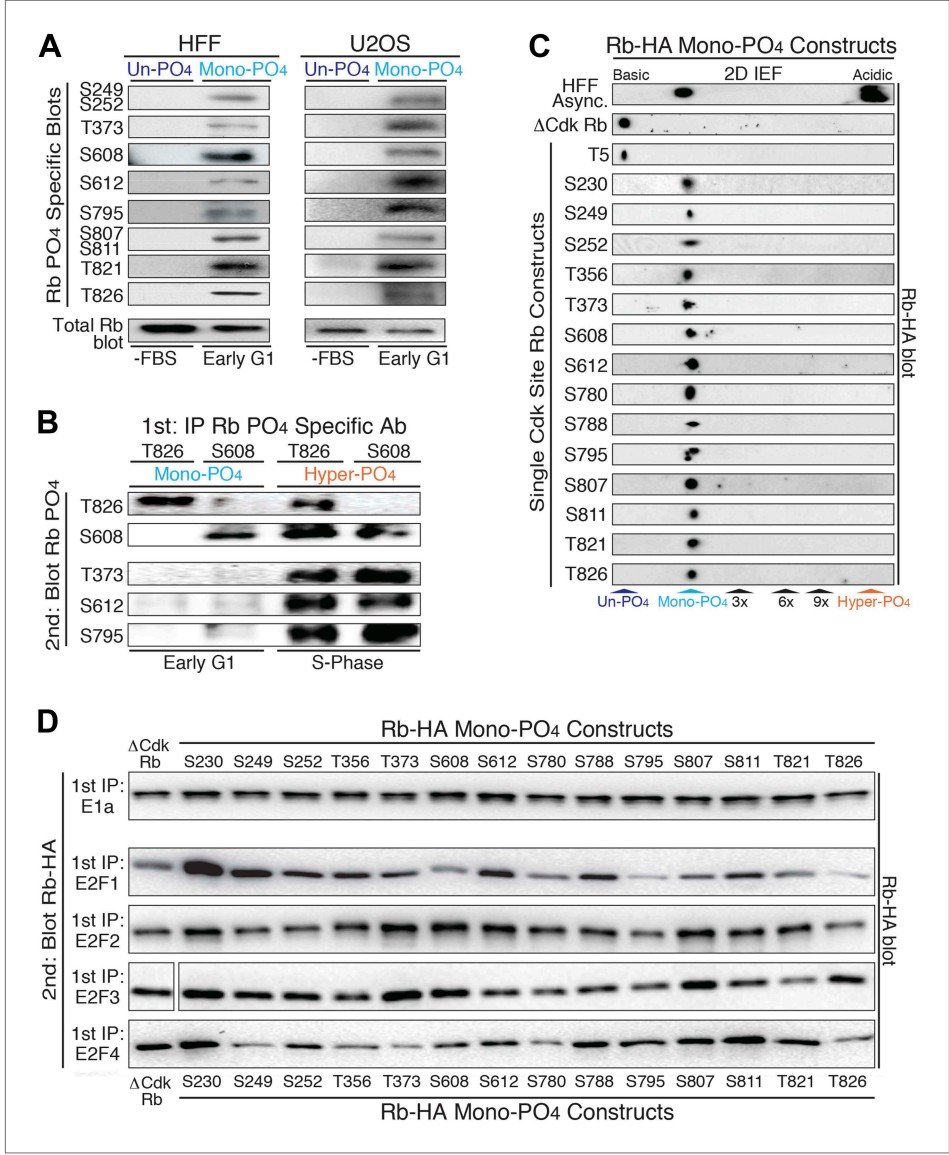

**Figure 2**. Mono-phosphorylated Rb exists as fourteen individual isoforms. (**A**) Phospho-specific Rb immunoblot of un-phosphorylated Rb ($G_0$, −FBS) and mono-phosphorylated Rb (contact inhibited early $G_1$, +FBS) from HFF and U2OS cells. (**B**) T826 and S608 phospho-specific Rb immunoprecipitation of mono-phosphorylated Rb (contact inhibited early $G_1$, +FBS) and hyper-phosphorylated Rb (S phase) from HFF cells, followed by phospho-specific Rb immunoblot analysis, as indicated. Note the absence of other phosphates at these locations on mono-phosphorylated Rb, but present on hyper-phosphorylated Rb. (**C**) 2D IEF Rb-HA immunoblot of single Cdk site Rb-HA constructs expressed in cycling cells. Numbering indicates single Cdk site location on Rb. (**D**) Immunoblot of Rb-HA single Cdk site constructs and control ΔCdk Rb construct from co-immunoprecipitated and co-expressed E1a, E2F1, E2F2, E2F3 or E2F4 (Myc tagged) as indicated. Numbering indicates single Cdk site location on Rb. Note that all 14 avidly bind to at least two E2F family members and there are no completely inactive mono-phosphorylated Rb isoforms.

The following figure supplements are available for figure 2:

**Figure supplement 1**. Mono-Phosphorylated Rb exists as fourteen individual isoforms.

Surprisingly, we found that none of the mono-phosphorylated Rb isoforms were completely inactive for binding to E2Fs and that each E2F showed a preferential binding specificity for individual mono-phosphorylated Rb isoforms (*Figure 2D*). While E2F2 and E2F3 showed subtle biases for specific mono-phosphorylated Rb isoforms, E2F1 and E2F4 showed the largest enhanced or decreased

binding specificities to each Rb mono-phosphorylated isoform. E2F1 showed enhanced binding to Rb when it was mono-phosphorylated at positions S230, S249, T356 and S612 and decreased avidity when Rb was mono-phosphorylated at positions S608, S795 and T826 (*Figure 2D*). E2F1 bound all of the other mono-phosphorylated Rb isoforms with comparable avidity as control un-phosphorylated ΔCdk Rb. E2F4 showed enhanced binding when Rb was mono-phosphorylated at positions S230, S788 and S811, and decreased binding when it was mono-phosphorylated at positions S249, T373, S780 and T826 (*Figure 2D*). While phosphorylation of T373 has recently been singled out as an inactivating phosphorylation on a fragment of Rb (*Burke et al., 2012*), in our hands E1a, E2F1, E2F2, and E2F3 all bound T373 mono-phosphorylated Rb when the full-length protein was expressed in cells. Together, these observations demonstrated the presence of 14 independent mono-phosphorylated Rb isoforms that are present in early $G_1$ phase and showed that each has differential binding preferences to E2F family members. These results parallel other signaling proteins where phosphorylation of specific sites enhance or decrease binding to cellular targets.

## Cyclin D:Cdk4/6 complexes exclusively mono-phosphorylate Rb in early $G_1$ phase

Based on the constitutive cyclin D:Cdk4/6 activity in early $G_1$ phase when Rb was exclusively mono-phosphorylated (*Figure 1B*), cyclin D:Cdk4/6 became a prime candidate for the Rb mono-phosphorylating kinase. To dissect the role of cyclin D:Cdk4/6 to phosphorylate Rb, we used triple knockout (TKO) cyclin D genetic deletion in mouse embryonic fibroblasts (MEFs) (*Choi et al., 2012*). MEFs containing a deleted cyclin D2 gene and homozygous LoxP cyclin D1$^{f/f}$ and D3$^{f/f}$ genes were treated with Adenovirus Cre recombinase to generate TKO cyclin D$^-$ MEFs. TKO cyclin D$^-$ MEFs continuously cycled and contained hyper-phosphorylated Rb by 1D SDS-PAGE (*Figure 3A*, *Figure 3— figure supplement 1*). 2D IEF of Rb from asynchronously cycling parental D1$^+$/D3$^+$ MEFs showed the presence of both mono-phosphorylated Rb and hyper-phosphorylated Rb isoforms (*Figure 3B*). In contrast, 2D IEF of Rb from cycling TKO cyclin D$^-$ MEFs contained un-phosphorylated Rb and hyper-phosphorylated Rb, with no mono-phosphorylated Rb detected (*Figure 3B*). Moreover, contact inhibited early $G_1$ phase arrested (+FBS) parental D1$^+$/D3$^+$ MEFs contained only mono-phosphorylated Rb, whereas contact inhibited (+FBS) TKO cyclin D$^-$ MEFs contained only un-phosphorylated Rb (*Figure 3B*). Retroviral expression of cyclin D1 in contact inhibited TKO cyclin D$^-$ MEFs resulted in the appearance of mono-phosphorylated Rb (*Figure 3B*).

To further test the notion that cyclin D:Cdk4/6 is the Rb mono-phosphorylating kinase, we treated serum deprived $G_0$ arrested (−FBS) and restimulated (+FBS) HFFs to enter early $G_1$ phase with a selective Cdk4/6 inhibitor (PD0332991) (*Fry et al., 2004*). Consistent with TKO cyclin D$^-$ MEFS, treatment of HFFs with the Cdk4/6 inhibitor at restimulation (+FBS) resulted in the presence of un-phosphorylated Rb, whereas control DMSO-treated cells contained mono-phosphorylated Rb (*Figure 3C*). Likewise, specific inhibition of cyclin D:Cdk4/6 complexes by retroviral expression of the Cdk4/6 inhibitor p16 resulted in the presence of un-phosphorylated Rb (*Figure 3C*). Furthermore, induction of p16 in serum deprived $G_0$ arrested (−FBS) and restimulated (+FBS) U2OS cells (*Jiang et al., 1998*) also resulted in the presence of un-phosphorylated Rb, whereas control cells (repressed p16) contained mono-phosphorylated Rb (*Figure 3D*). Collectively, these results from three independent approaches (genetic TKO cyclin D$^-$ MEFs, expression of p16, treatment with Cdk4/6 inhibitor) in three different cell types (MEFs, HFFs, U2OS) confirmed that cyclin D:Cdk4/6 was the Rb mono-phosphorylating kinase in vivo.

Rb remains hyper-phosphorylated in M phase and is dephosphorylated by activation of the PP1 phosphatase as cells exit mitosis into the next early $G_1$ phase (*Ludlow et al., 1993*). To ascertain if phosphatase activity dephosphorylates Rb to mono-phosphorylated or un-phosphorylated isoforms, we arrested U2OS cells at $G_2$/M phase by addition of nocodazole, a microtubule depolymerizer, and detected only hyper-phosphorylated Rb (*Figure 3E,F*). After nocodazole washout and release of U2OS cells into early $G_1$ phase for 4 hr, all of the hyper-phosphorylated Rb was converted to mono-phosphorylated Rb. However, release of U2OS cells from the nocodazole arrest in the presence of the Cdk4/6 inhibitor (PD0332991) resulted in the exclusive appearance of un-phosphorylated Rb (*Figure 3E,F*). Likewise, nocodazole block and release of HeLa cells in the presence of the Cdk4/6 inhibitor (PD0332991) resulted in the presence of only un-phosphorylated Rb (*Figure 3G,H*). These results determined that as cells exit mitosis, Rb is fully dephosphorylated to un-phosphorylated Rb, and then rapidly mono-phosphorylated by cyclin D:Cdk4/6 complexes in early $G_1$ phase.

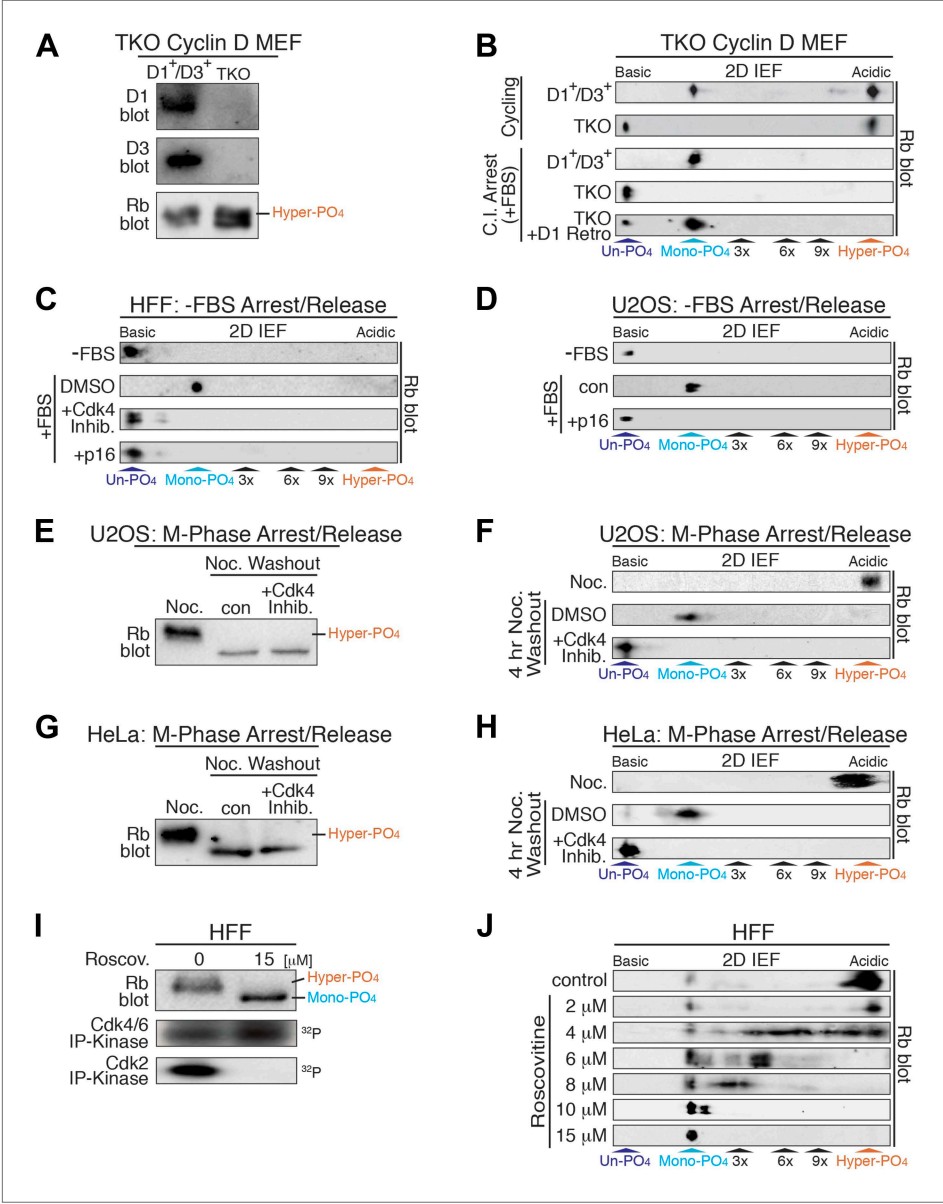

**Figure 3**. Cyclin D:Cdk/6 is the Rb mono-phosphorylation kinase. (**A**) Immunoblot of cyclin D1, D3, Rb in cycling parental D1+/D3+ MEFs and triple knockout (TKO) cyclin D− MEFs. (**B**) 2D IEF Rb immunoblot from cycling parental D1+/D3+ MEFs and TKO cyclin D− MEFs, and contact inhibited early $G_1$ phase arrested (+FBS) parental D1+/D3+ MEFs and TKO cyclin D− MEFs plus/minus retroviral cyclin D1 expression. (**C**) 2D IEF Rb immunoblot from serum deprived $G_0$ arrested (−FBS) and released (+FBS) HFFs plus control DMSO, Cdk4 inhibitor (PD0332991) or retroviral p16 expression. (**D**) 2D IEF Rb immunoblot from serum deprived $G_0$ arrested (−FBS) and released (+FBS) p16-deleted U2OS tumor cells or released (+FBS) plus TET-induced p16 expression. (**E**, **F**) 1D SDS-PAGE (**E**) and 2D IEF (**F**) Rb immunoblot of $G_2$/M phase nocodazole arrested (Noc.) and released U2OS cells plus DMSO (con) or Cdk4 inhibitor (PD0332991). (**G, H**) 1D SDS-PAGE (**G**) and 2D IEF (**H**) Rb immunoblot of $G_2$/M phase nocodazole arrested (Noc.) and released HeLa cells plus DMSO (con) or Cdk4 inhibitor (PD0332991). (**I**) Late $G_1$ phase primary HFFs were treated with Cdk2 inhibitor Roscovitine [15 μM] or control (DMSO) and analyzed by 1D SDS-PAGE Rb immunoblot, anti-Cdk4/6 and anti-Cdk2 immunoprecipitation-kinase assays. (**J**) 2D IEF Rb immunoblot from late $G_1$ phase HFFs treated with dose curve of Cdk2 inhibitor Roscovitine or control (DMSO).

The following figure supplements are available for figure 3:

**Figure supplement 1**. Cyclin D:Cdk/6 is the Rb mono-phosphorylation kinase.

Previous studies from our lab and many others have shown that Rb becomes inactivated by hyper-phosphorylation at the late $G_1$ Restriction point and remains hyper-phosphorylated throughout late $G_1$ phase, S phase, $G_2$ phase, and M phases (*DeCaprio et al., 1989*; *Mittnacht et al., 1994*; *Ezhevsky et al., 2001*). Hyper-phosphorylated Rb first appears concomitant with activation of cyclin E:Cdk2 complexes (*Figure 1B*). To ascertain the role of cyclin E:Cdk2 as the initial Rb hyper-phosphorylating kinase, we analyzed Rb from contact inhibited and released HFFs that were allowed to enter late $G_1$ phase at 12 hr. We found that the vast majority of Rb was hyper-phosphorylated, with active Cdk2 and active Cdk4/6 complexes (*Figure 3I*). Selective inhibition of Cdk2 by 15 μM roscovitine, a Cdk2 ATP competitive inhibitor, resulted in no Cdk2 activity, continued Cdk4/6 activity and the presence of mono-phosphorylated Rb (*Figure 3I,J*). Titration of roscovitine from 2 μM to 15 μM resulted in a dose-dependent appearance of intermediate phosphorylated Rb isoforms (*Figure 3J*), suggesting that unlike cyclin D:Cdk4/6, cyclin E:Cdk2 is a processive Rb kinase. These observations are entirely consistent with our previous reports that cyclin E:Cdk2 complexes are the initial Rb hyper-phosphorylating kinase at the late $G_1$ Restriction Point (*Ezhevsky et al., 1997*, *2001*; *Haberichter et al., 2007*).

## Mono-phosphorylated Rb is functionally active during a DNA damage response

Although Rb regulates many processes in early $G_1$ phase (*Burkhart and Sage, 2008*), to ascertain if mono-phosphorylated Rb was functionally active, we focused on Rb's regulation of a DNA damage response cell cycle arrest (*Harrington et al., 1998*; *Brugarolas et al., 1999*; *Knudsen et al., 2000*; *Avni et al., 2003*). Treatment of cycling MEFs with a sub-lethal dose (100 ng/ml) of doxorubicin, a DNA damage-inducing topoisomerase II inhibitor, resulted in a $G_1$ phase cell cycle arrest with constitutive cyclin D:Cdk4/6 activity, mono-phosphorylated Rb, and loss of cyclin E/A:Cdk2 activity (*Figure 4A,B*). Surprisingly, doxorubicin treatment of serum-deprived $G_0$ arrested MEFs that contained un-phosphorylated Rb and no cyclin D:Cdk4/6 activity, resulted in induction of cyclin D1 and Cdk6, activation of cyclin D:Cdk4/6 complexes and Rb mono-phosphorylation (*Figure 4C,D*). Although several studies have suggested that Rb is phosphorylated during a DNA damage response by non-Cdk kinases, including Chk1/2 and Aurora B (*Inoue et al., 2007*; *Nair et al., 2009*), inhibition of Cdk4/6 activity by retroviral p16 expression in doxorubicin-treated MEFs resulted in the exclusive presence of un-phosphorylated Rb (*Figure 4B,D*), thereby excluding the involvement of other kinases.

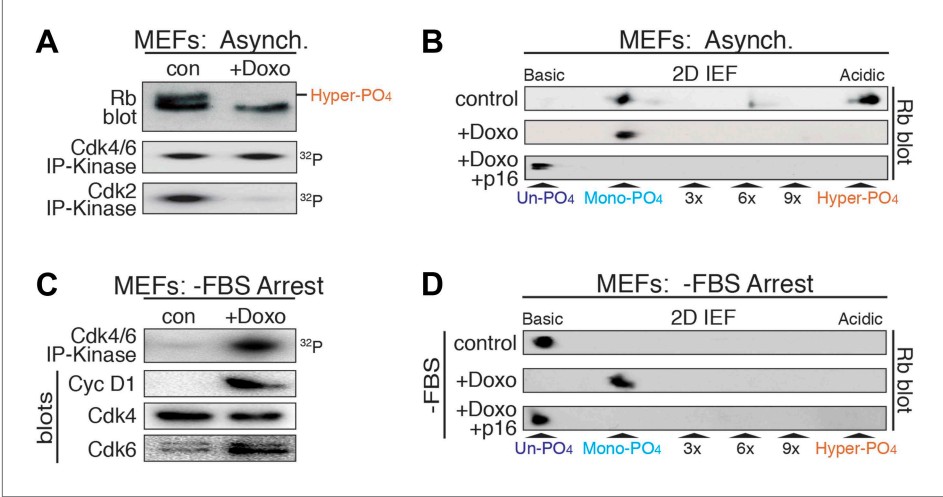

**Figure 4**. DNA damage induces cyclin D:Cdk4/6 activity and mono-phosphorylated Rb. (**A**) Asynchronously cycling MEFs (+FBS) treated with Doxorubicin (+Doxo, 100 ng/ml) were analyzed by 1D SDS-PAGE Rb immunoblot, anti-Cdk4/6 and anti-Cdk2 immunoprecipitation-kinase assays. (**B**) 2D IEF Rb immunoblot from asynchronously cycling MEFs (+FBS) treated with control or Doxorubicin (+Doxo) or Doxorubicin plus retroviral p16 (+Doxo/+p16) expression. (**C**) Serum-deprived $G_0$ arrested MEFs (−FBS) treated with Doxorubicin (+Doxo/−FBS) were immunoblot analyzed for cyclin D1, Cdk4, Cdk6, and Cdk4/6 immunoprecipitation-kinase activity. Note activation of cyclin D:Cdk4/6 in absence of serum growth factors. (**D**) 2D IEF Rb immunoblot from serum-deprived $G_0$ arrested MEFs (−FBS) treated with control or Doxorubicin (+Doxo) or Doxorubicin plus retroviral p16 (+Doxo/+p16) expression.

Together, these observations demonstrated that in response to DNA damage, cells select for mono-phosphorylated Rb by activating cyclin D:Cdk4/6 complexes.

We developed a genetic system to test if un-phosphorylated Rb and/or mono-phosphorylated Rb were functionally active to mediate the DNA damage cell cycle arrest checkpoint. Parental homozygous LoxP Rb$^{f/f}$ MEFs (*Marino et al., 2000*) were treated with TAT-Cre protein (*Wadia et al., 2004*) to delete Rb (Rb$^{-/-}$), followed by infection with carefully engineered and titered wild type (WT) Rb-HA or non-phosphorylatable ΔCdk Rb-HA retroviruses (*Figure 5A*). This approach resulted in an acute loss of endogenous Rb protein combined with simultaneous replacement by ectopic Rb at physiologic levels (*Figure 5A*). Treatment of Rb-deleted MEFs expressing ectopic WT Rb-HA with doxorubicin resulted in the presence of mono-phosphorylated Rb, whereas Rb-deleted MEFs expressing ectopic ΔCdk Rb-HA contained un-phosphorylated Rb (*Figure 5B*).

We next analyzed the ability of WT Rb-HA and ΔCdk Rb-HA to regulate global transcription during an acute DNA damage response. Parental MEFs expressing endogenous Rb, Rb-deleted MEFS, and Rb-deleted MEFS expressing either WT Rb-HA or ΔCdk Rb-HA were contact arrested in early G$_1$ phase (+FBS) and released for 4 hr, then treated with a sub-lethal dose of doxorubicin (100 ng/ml) for 3 hr and analyzed for whole genome transcriptional changes (*Figure 5C*). Rb is a transcriptional repressor and Rb-deleted MEFs showed a >1.6x increase in expression of 173 genes, primarily involved in DNA replication (24%), cell cycle control (20%), and regulation of transcription (18%) (*Table 1*). Expression of physiologic levels of mono-phosphorylated WT Rb-HA in Rb-deleted cells restored repression of many of these genes, especially E2F target genes (*Figure 5C,D*; *Figure 5—figure supplement 1*). However, expression of un-phosphorylated ΔCdk Rb-HA at physiologic levels failed to repress genes and gave a pattern of global transcriptional deregulation similar to Rb-deleted MEFs, suggesting that un-phosphorylated Rb was functionally inactive during a DNA damage response checkpoint during early G$_1$ phase. Further qRT-PCR analysis of two strong E2F target genes, DHFR and cdc6, showed similar levels of deregulation in both Rb-deleted and un-phosphorylated ΔCdk Rb-HA MEFs, whereas WT Rb-HA repressed both of these genes (*Figure 5E–G*). A non-Rb regulated gene, p21, showed no difference between all three genotypes. Rb-deleted (Rb$^{-/-}$) MEFs also failed to prevent the appearance of tetraploid cells several days after treatment with either doxorubicin or ionizing radiation (20 Grays) (*Figure 5H,I*). Consistent with the inability to regulate transcriptional control, un-phosphorylated ΔCdk Rb-HA MEFs showed similar high levels of tetraploid cells in response to DNA damage as Rb-deleted (Rb$^{-/-}$) MEFs. In contrast, WT Rb-HA expression rescued the tetraploid phenotype to levels near parental MEFs expressing endogenous wild type Rb (*Figure 5H,I*). These observations demonstrated that cells undergoing a DNA damage response activate cyclin D:Cdk4/6 to generate biologically active, mono-phosphorylated Rb, whereas un-phosphorylated was functionally inactive for regulating E2F transcription and preventing the appearance of tetraploid cells.

## Un-phosphorylated Rb promotes cell cycle exit and differentiation

The above results demonstrated that un-phosphorylated Rb was non-functional during a DNA damage response. However, serum-deprived G$_0$ arrested HFFs and U2OS cells contained un-phosphorylated Rb (*Figure 1E,G*; *Figure 3D*), as do quiescent G$_0$ peripheral blood lymphocytes (PBLs) (*Ezhevsky et al., 2001*). Moreover, other studies have documented a role for Rb in cell cycle exit and differentiation, including proper myogenic development as myoblasts exit the cell cycle into G$_0$ (*Gu et al., 1993*; *Zacksenhaus et al., 1996*; *Chen and Wang, 2000*; *Sage et al., 2003*; *Blais et al., 2007*). Together, these observations suggested a potential functional role for un-phosphorylated Rb during cell cycle exit and differentiation. To evaluate the phosphorylation status of Rb during differentiation, we used the well-established C2C12 myoblast to myotube differentiation system (*Blais et al., 2005*). Asynchronous cycling C2C12 myoblasts grown in high mitogen media (FBS) contained both mono-phosphorylated Rb and hyper-phosphorylated Rb with no detectable un-phosphorylated Rb (*Figure 6A,B*). However, addition of low mitogen, differentiation media induced expression of myotube specific myogenin and resulted in the exclusive presence of un-phosphorylated Rb at day 2, concomitant with loss of cyclin D:Cdk4/6 kinase activity and expression of p18, a Cdk4/6-specific inhibitor (*Halevy et al., 1995*; *Franklin and Xiong, 1996*; *Wang and Walsh, 1996*; *Zhang et al., 1999*) (*Figure 6B–D*). We also observed the exclusive appearance of un-phosphorylated Rb when human HL60 promyelocytic cells were induced to undergo differentiation by addition of retinoic acid (*Figure 6—figure supplement 1A,B*).

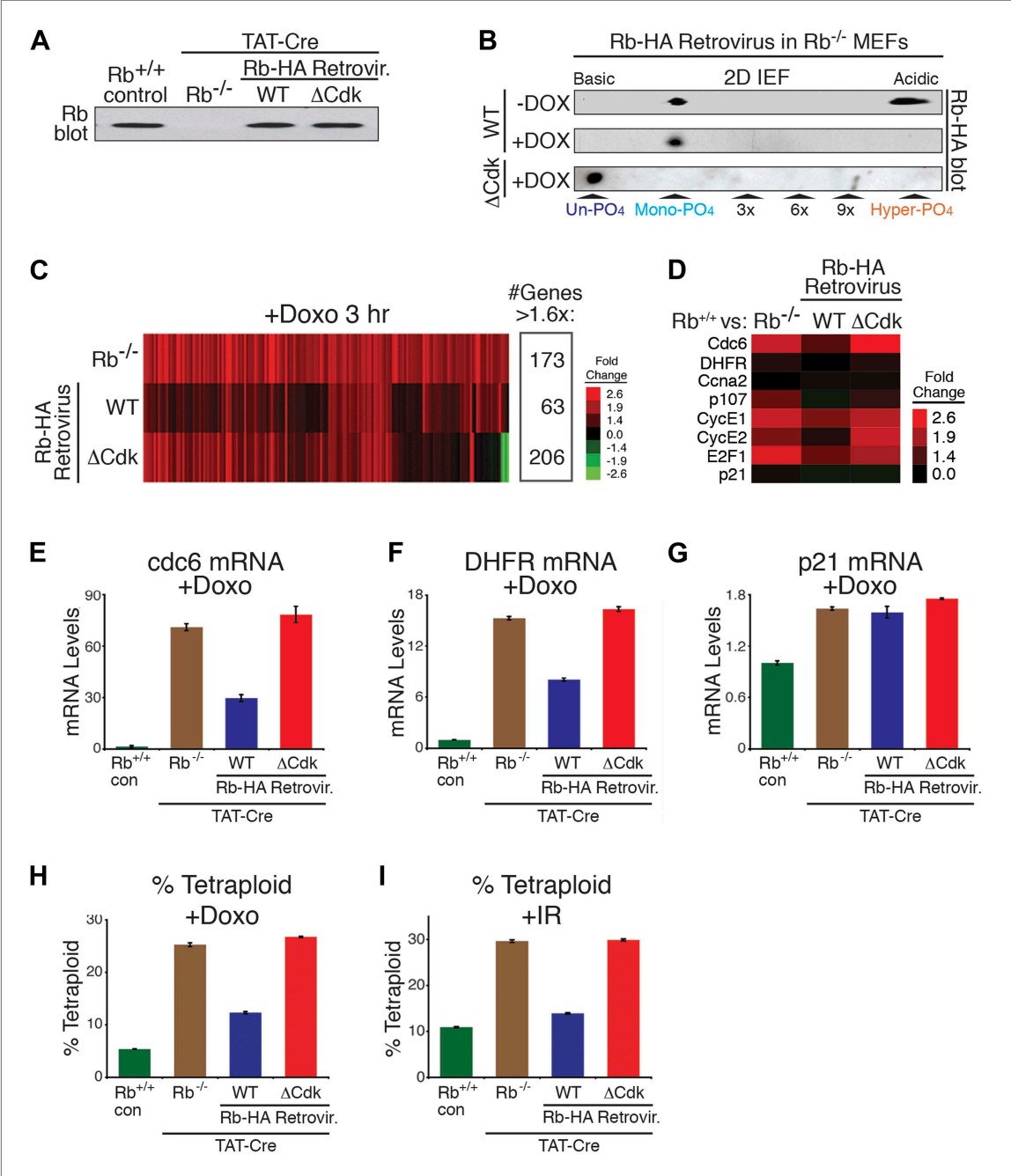

**Figure 5**. Mono-phosphorylated Rb is active during a DNA damage response. (**A**) Rb immunoblot of contact inhibited parental MEFs, conditionally deleted Rb$^{-/-}$ MEFs, and retrovirally-expressed wild type (WT) Rb-HA or non-phosphorylatable ΔCdk Rb-HA in deleted Rb$^{-/-}$ MEFs. (**B**) 2D IEF Rb immunoblot of cycling and doxorubicin (+Dox) (100 ng/ml) treated WT Rb-HA MEFs, and ΔCdk Rb-HA MEFs in deleted Rb$^{-/-}$ background. (**C**) Microarray heat map of mRNA levels from early G$_1$ phase Rb$^{-/-}$ MEFs, WT Rb-HA MEFs, and ΔCdk Rb-HA MEFs compared to parental MEFs treated with Doxorubicin (+Doxo) for 3 hr. Only genes increased >1.6-fold are shown. (**D**) Focused mRNA expression analysis of E2F target genes and p21 control gene from (**C**). (**E–G**) qRT-PCR mRNA analysis of endogenous E2F-dependent target genes, cdc6 (**E**) and DHFR (**F**), and a non-E2F control gene, p21 (**G**), from parental MEFs (con), Rb$^{-/-}$ MEFs, WT Rb-HA MEFs, and ΔCdk Rb-HA MEFs in deleted Rb$^{-/-}$ background treated with doxorubicin (+Doxo). Mean values were normalized to β2-microglobulin levels, reported as fold change from parental MEFs (con). Error bars indicate SEM from three independent experiments. (**H** and **I**) Quantification of percent tetraploid (>4n DNA) nuclei in parental MEFs (con), Rb$^{-/-}$ MEFs, WT Rb-HA MEFs, and ΔCdk Rb-HA MEFs in deleted Rb$^{-/-}$ background four days after treatment with Doxorubicin (+Doxo) (**H**) or 2 days post-treatment with ionizing radiation (20 grays) (**I**). Error bars indicate SEM from three independent experiments.

The following figure supplements are available for figure 5:

**Figure supplement 1**. Mono-phosphorylated Rb is active during a DNA damage response.

**Table 1.** Mono-phosphorylated Rb is active during a DNA damage response

| Induced | | Repressed | |
| --- | --- | --- | --- |
| DNA replication | 24% | Localization | 18% |
| Cell cycle | 20% | Post-translational protein modification | 10% |
| Regulation of transcription | 18% | Biosynthetic process | 9% |
| Organelle organization and biogenesis | 12% | Organelle organization and biogenesis | 8% |
| Response to stress | 12% | Intracellular signaling cascade | 8% |
| DNA repair | 8% | Proteolysis | 6% |
| DNA packaging | 7% | Catabolic process | 6% |
| Macromolecular complex assembly | 6% | Cell cycle | 6% |
| Cellular component assembly | 6% | Nervous system development | 6% |
| Chromatin assembly | 6% | Response to external stimulus | 4% |
| | | Response to wounding | 3% |
| | | Dephosphorylation | 2% |

Gene ontology of mRNA differences showing a >1.6-fold increase/decrease level by microarray analysis between parental MEFs and Rb$^{-/-}$ MEFs treated with 100 ng/ml doxorubicin for 3 hr after release from contact inhibition.

To directly test for a role of un-phosphorylated Rb in regulating cell cycle exit and differentiation in myoblasts, we devised a strategy similar to the MEF DNA damage response approach used above. Endogenous Rb from C2C12 myoblasts was knocked down by shRb or control shScramble (Scr) RNAi retroviruses, followed by retroviral physiologic expression of WT Rb-HA or ΔCdk Rb-HA (*Figure 6E*). As per the method of *Blais et al. (2007)*, the efficiency of cell cycle exit was evaluated 2 days post-addition of low mitogen differentiation media by counting the number of nuclei (*Figure 6F–H*). Reduction of Rb levels by shRNAi resulted in an increased proliferation index (number of nuclei), and increased expression of Mcm3 and Mcm5 (markers of DNA replication) (*Figure 6F–H*, *Figure 6—figure supplement 1C*). However, physiologic expression of WT Rb-HA compensated for the reduction in endogenous Rb by bringing the proliferation index back to the shScrambled RNAi control. Strikingly, physiologic expression of ΔCdk Rb-HA resulted in a dramatic reduction of proliferation as cells prematurely exited the cell cycle and decreased Mcm3 and Mcm5 levels (*Figure 6F–H*; *Figure 6—figure supplement 1C*). Thus, in contrast to the DNA damage response where un-phosphorylated Rb was non-functional, these observations demonstrated that un-phosphorylated Rb was functionally active to drive cell cycle exit and differentiation of myoblasts into myotubes.

## Discussion

For the last 20 years, the key tenet of the prevailing model of G$_1$ cell cycle progression proposed that cyclin D:Cdk4/6 complexes inactivated Rb by progressive multi-phosphorylation, termed hypo-phosphorylation, resulting in the gradual release of E2F transcription factors that drive cells into late G$_1$ phase. This notion was reinforced when tumor cells expressing wild-type Rb were found to have genetic and epigenetic alterations of the p16 tumor suppressor gene or oncogenic expression of cyclin D1, D2, D3, Cdk4, and Cdk6 genes, which became known as the 'p16-cyclin D-Rb' pathway (*Sherr, 1994*; *Sherr and McCormick, 2002*; *Burkhart and Sage, 2008*; *Paternot et al., 2010*; *Henley and Dick, 2012*; *Choi and Anders, 2013*). Further reinforcing this notion were experiments utilizing supra-physiologic overexpression of D-type cyclins and Cdk4/Cdk6 that inactivated Rb and drove cells into S phase. Likewise, experiments overexpressing p16, a Cdk4/6 inhibitor, arrested cells in a 2n DNA content and were interpreted to confirm the notion that cyclin D:Cdk4/6 complexes inactivated Rb. Moreover, the large number of tryptic phospho-peptides of 'hypo-phosphorylated' Rb from early G$_1$ cells reported by *Mittnacht et al. (1994)* was unknowingly misinterpreted and further reinforced the notion that cyclin D:Cdk4/6 complexes inactivate Rb by progressive multi-phosphorylating, hypo-phosphorylation. Although there was a complete absence of rigorous biochemical evidence as to the extent of phosphate numbers or kinetics on what was loosely termed hypo-phosphorylated Rb, the

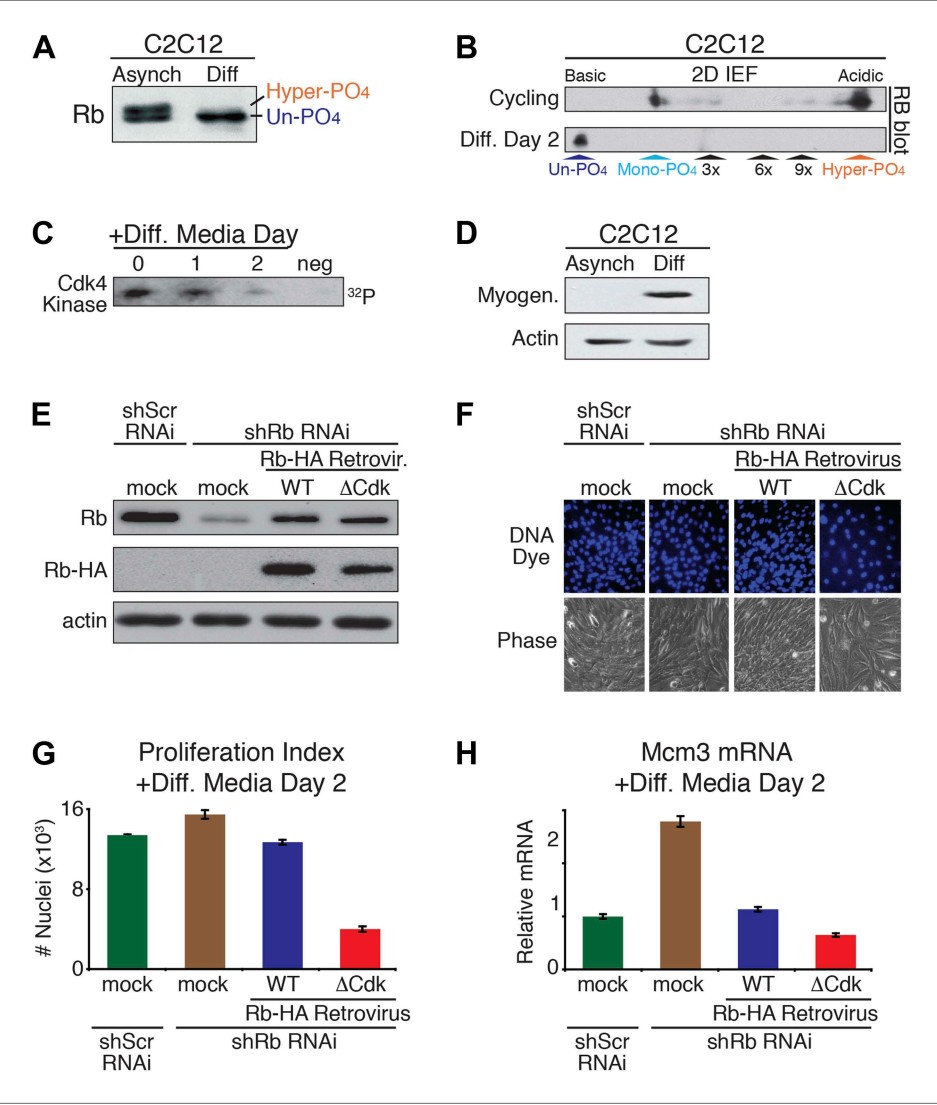

**Figure 6**. Un-phosphorylated Rb promotes cell cycle exit and differentiation. (**A**) Rb immunoblot of cycling C2C12 myoblasts (Asynch), and after 2 days in differentiation medium (Diff). (**B**) 2D IEF Rb immunoblot from cycling C2C12 myoblasts (Asynch) and after 2 days in differentiation medium (Diff). (**C**) Cdk4/6 immunoprecipitation-kinase assay of cycling C2C12 myoblasts (0) and at 1 and 2 days post-addition of differentiation medium (Diff). Negative (neg) control, irrelevant antibody. (**D**) Myogenin immunoblot of cycling C2C12 myoblasts (Asynch), and after 2 days in differentiation medium (Diff). (**E**) Immunoblot of endogenous Rb, and retroviral expressed wild type (WT) Rb-HA or ΔCdk Rb-HA in C2C12 myoblasts co-infected with short hairpin (sh) RNAi retroviruses targeting endogenous Rb (3' UTR) or scrambled (Scr) control. (**F** and **G**) Proliferation analysis (number of nuclei) of C2C12 myoblasts co-infected with short hairpin (sh) RNAi retroviruses targeting endogenous Rb (3' UTR) or scrambled (Scr) control, and wild type (WT) Rb-HA or ΔCdk Rb-HA retroviruses. Cells were stained with Hoechst 33342 DNA dye, visualized by microscopy (**F**) and quantified by flow cytometry (**G**) (number of nuclei x $10^3$) at 2 days post-addition of differentiation medium. Error bars indicate SEM. (**H**) qRT-PCR mRNA analysis of Mcm3 DNA replication factor in C2C12 myoblasts treated as above (**G**) for 2 days in differentiation media. Mean values were normalized to β2-microglobulin levels and reported as fold change from C2C12 myotubes expressing endogenous Rb (Scr[shRNA]). Error bars indicate SEM from three independent experiments.

The following figure supplements are available for figure 6:

**Figure supplement 1**. Un-phosphorylated Rb promotes cell cycle exit and differentiation.

notion that cyclin D:Cdk4/6 inactivated Rb by progressive multi-phosphorylating, hypo-phosphorylation was solidified in the 1990s as the model of $G_1$ cell cycle progression.

In sharp contrast to the prevailing model, we had previously performed kinetic analyses from highly synchronized normal cells and p16-deficient tumor cells, and found that cyclin D:Cdk4/6 was constitutively active throughout all of the early $G_1$ phase at the same time points that Rb was actively binding E2Fs and repressing E2F target gene expression (*Ezhevsky et al., 1997*, *2001*; *Haberichter et al., 2007*). Consequently, we suspected that cyclin D:Cdk4/6 complexes were not inactivating Rb, but may, in fact, be activating Rb by phosphorylation. The key to understanding the relationship between Rb and cyclin D:Cdk4/6 complexes was developing the ability to quantitatively separate all Rb isoforms by 2D IEF combined with the generation of Rb phosphorylation standards. During the course of this study, we performed hundreds of 2D IEFs on Rb from 11 independent normal and tumorigenic cell types under conditions of asynchronous cycling cells, cells arrested in $G_0$, early $G_1$, late $G_1$ or $G_2$/M phases, and cells that were arrested/released and followed kinetically. Under all of these conditions and cell types, we found that Rb was exclusively mono-phosphorylated throughout all of early $G_1$ phase in both normal and tumor cells, and hyper-phosphorylated in late $G_1$, S, $G_2$, and M phases. In fact, we found no biochemical evidence to support the prevailing $G_1$ cell cycle model that cyclin D:Cdk4/6 progressively hypo-phosphorylates Rb. Moreover, using three independent approaches to dissect cyclin D:Cdk4/6 function on Rb, namely: triple cyclin D genetic deletion, addition of a Cdk4/6-specific chemical inhibitor, and p16 expression, we determined that cyclin D:Cdk4/6 is the Rb mono-phosphorylating kinase (*Figure 7*). Given that mono-phosphorylated Rb is the only isoform of Rb present in early $G_1$ phase, by definition, some, most or all of the mono-phosphorylated Rb isoforms must be biologically active. Consequently, it was not too surprising that mono-phosphorylated Rb was the active Rb isoform mediating a DNA damage response cell cycle arrest and regulating global transcription. However, we note the unanticipated observation that exposure of quiescent $G_0$ (−FBS) primary cells, containing

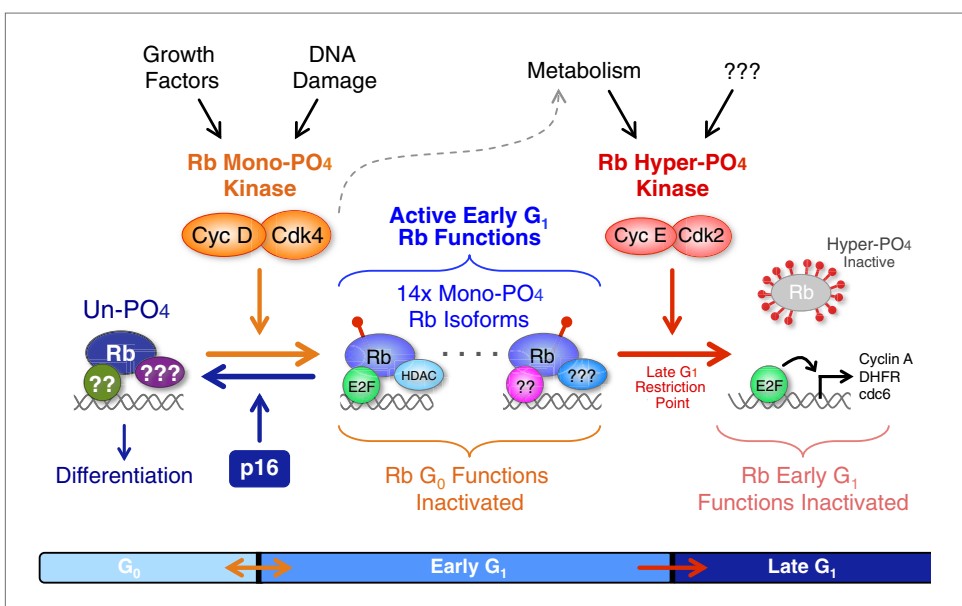

**Figure 7**. Revised working model of $G_1$ cell cycle progression. Un-phosphorylated Rb regulates $G_0$ cell cycle exit and differentiation. Growth factor signaling and DNA damage stimulate activation of cyclin D:Cdk4/6 complexes that diversify Rb into 14 mono-phosphorylated isoforms that independently bind specific cellular factors to regulate early $G_1$ phase functions and the DNA damage response. Cyclin D:Cdk4/6 mono-phosphorylation of Rb inactivates un-phosphorylated Rb $G_0$ functions and thereby prevents cells from exiting the cell cycle. Activation of cyclin E:Cdk2 complexes inactivates all 14 mono-phosphorylated Rb isoforms by hyper-phosphorylation (>12x phosphates) at the late $G_1$ Restriction Point. Cyclin A:Cdk2 and cyclin B:Cdk1 maintain Rb in an inactive hyper-phosphorylated state during S, $G_2$ and M phases. As cells complete cytokinesis, hyper-phosphorylated Rb is de-phosphorylated by phosphatases and rapidly mono-phosphorylated by cyclin D:Cdk4/6 complexes. We speculate that an unknown metabolic sensor is upstream of cyclin E:Cdk2 activation.

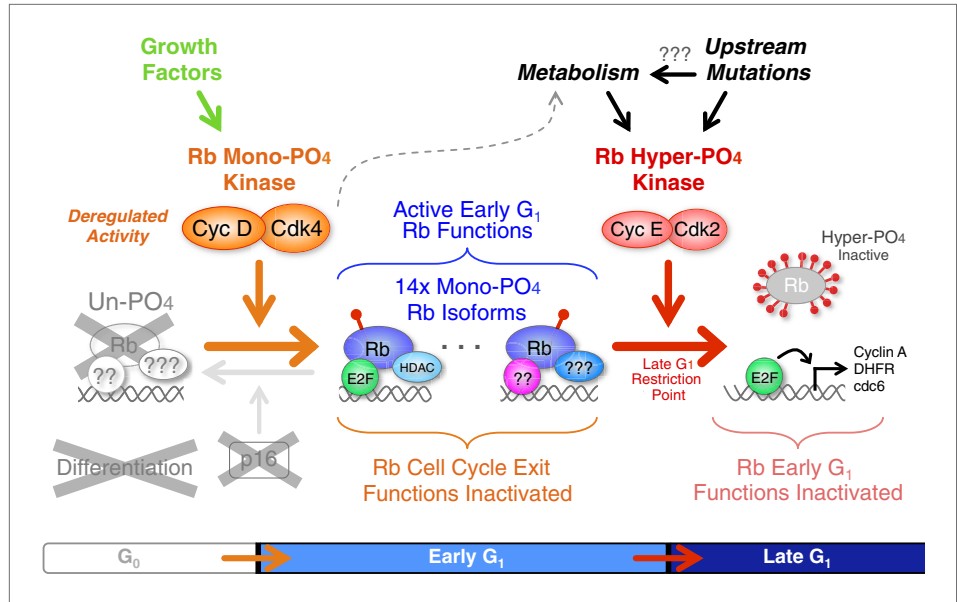

**Figure 8**. Deregulated cyclin D:Cdk4/6 in cancer mono-phosphorylates Rb to prevent cell cycle exit. Deregulation of cyclin D:Cdk4/6 activity in cells occurs by a variety of mechanisms, including: p16 deletion, cyclin D1, D2 and D3 amplification or overexpression, and mutation or overexpression of Cdk4 or Cdk6. Cyclin D:Cdk4/6 mono-phosphorylation of Rb simultaneously inactivates Rb's $G_0$ functions and activates Rb's early $G_1$ phase functions thereby driving cells from a low metabolism $G_0$ quiescence into a high metabolism early $G_1$ arrested state that also prevents subsequent cell cycle exit or differentiation. Similar to p53 mutations and Bcl2 overexpression, deregulated cyclin D:Cdk4/6 activity is a well tolerated priming oncogenic mutation that avoids activation of oncogene-induced apoptosis. The net effect is a subtle, but irreversible, oncogenic step forward. We predict that additional oncogenic and metabolic pathways ultimately converge on and activate cyclin E:Cdk2 complexes to inactivate Rb by hyper-phosphorylation at the Restriction Point and drive cells into late $G_1$ phase.

no cyclin D:Cdk4/6 activity, to DNA damaging agents induced and activated cyclin D:Cdk4/6 complexes to mono-phosphorylate Rb in the absence of serum growth factors. This observation is consistent with a role for cyclin D:Cdk4/6 complexes in the DNA damage response checkpoint (*Choi et al., 2012*), but also raises a cautionary concern for a potential increase of genomic DNA damage in the normal cells of patients being simultaneously treated with a cyclin D:Cdk4/6 inhibitor and a DNA damaging chemotherapy or ionizing radiation.

The Rb gene is transcribed and spliced into a single 4.7 kb mRNA composed of 27 coding exons that does not undergo appreciable alternative splicing and encodes a single 105 kDa protein (*Burkhart and Sage, 2008*). Although the complete structure of Rb has yet to be solved, several groups have recently solved the core structure as an intra-strand pseudo-dimer of dimers with the Cdk sites present on loops between structured regions or on the C-terminus abutting the B-box (*Burke et al., 2012*; *Lamber et al., 2013*) (*Figure 1A*). Based on the binding preferences of individual Rb mono-phosphorylated isoforms to specific E2F transcription factors, our data suggest that the generation of 14 mono-phosphorylated Rb isoforms may serve as a mechanism to post-translationally functionally diversify Rb in early $G_1$ phase from a single un-phosphorylated isoform in $G_0$. In addition to E2F transcription factors, Rb has also been shown to bind to over 100 additional cellular proteins (*Morris and Dyson, 2001*), leaving open the potential for differential binding preferences of mono-phosphorylated Rb to specific cellular targets. Excluding hyper-phosphorylated Rb, we were surprised at the complete absence of any multi-phosphorylated Rb isoforms. This raised the mechanistic question of how does cyclin D:Cdk4/6 complexes place one, and only one, phosphate on Rb while leaving the remaining 13 Cdk sites un-phosphorylated? While this will require extensive structural analyses beyond the scope of our study, we speculate that the substrate recognition of Rb by cyclin D's N-terminal LxCxE motif weak binding to Rb's pocket domain (*Dowdy et al., 1993*) vs cyclins E and A avid binding

to four C-terminal R/KxL substrate motifs (*Adams et al., 1999*) outside of the pocket likely serves as the defining mechanism between Rb mono-phosphorylation and Rb hyper-phosphorylation (*Figure 1A*). In this hypothesis, cyclin E/A:Cdk2's strong binding to the C-terminal tail of Rb would allow access to all 14 Cdk sites on Rb even when transcription factors and chromatin remodeling factors were bound to Rb's pocket and N-terminal binding sites. This also allows for a simultaneous switch-like inactivation of all 14 mono-phosphorylated Rb isoforms by one processive hyper-phosphorylation mechanism. We believe that the utilization of the same phosphorylation sites for the activation and inactivation of a protein, albeit with a >12-fold increased phosphate stoichiometry, is an unprecedented mechanism in the literature, but one that also likely applies to Rb-related genes p107 and p130.

Our study addresses several critical problems arising from numerous biochemical analyses of Rb phosphorylation going back more than 20 years. The results presented here fundamentally change the understanding of $G_1$ cell cycle regulation to show that cyclin D:Cdk4/6 activates Rb for binding cellular targets during early $G_1$ phase by generating 14 independent mono-phosphorylated Rb isoforms. Given that the majority of human tumors contain wild-type Rb, but select for deregulated cyclin D:Cdk4/6 activity (*Sherr and McCormick, 2002*; *Burkhart and Sage, 2008*; *Knudsen and Knudsen, 2006*; *Choi and Anders, 2013*), we hypothesize that the oncogenic activation of cyclin D:Cdk4/6 results in Rb mono-phosphorylation to drive quiescent $G_0$ cells into a more metabolically active, but early $G_1$ phase arrested phenotype (*Figure 8*). By constitutively mono-phosphorylating Rb, the nascent neoplastic cell avoids cell cycle exit and differentiation mediated by un-phosphorylated Rb, and also maintains a high level of metabolism. This notion is entirely consistent with the observed subtle and highly tolerated cancer predisposing mutations of p16 deletion and cyclin D overexpression in mouse models that avoid activation of oncogene-induced apoptosis (*Burkhart and Sage, 2008*). The net effect is a subtle, but irreversible, oncogenic step forward. While our study determined the role of cyclin D:Cdk4/6 in mono-phosphorylating Rb, it leaves wide open the question of what the rate-limiting switch-like mechanism is to activate cyclin E:Cdk2, the first domino in Rb inactivation. Cyclin D:Cdk4/6 activity combined with other signal transduction pathway mutations contributes to increased cellular metabolism that we speculate is monitored by an unknown metabolic sensor. Once the metabolic threshold has been exceeded, the sensor activates cyclin E:Cdk2 resulting in Rb inactivation by hyper-phosphorylation, induction of E2F target gene transcription and progression across the Restriction Point into late $G_1$ phase (*Haberichter et al., 2007*). We are currently investigating the mechanics of this putative mechanism and the identity of the metabolic sensor.

## Materials and methods

### Cell culture

Cells were $G_0$ arrested by serum deprivation for 5 days, followed by addition of 10% FBS. Cells were plated at high density in 10% FBS to contact inhibit arrest in early $G_1$ phase for 48 hr, followed by replating at low density in 10% FBS. DNA damage was induced by addition of 100 ng/ml doxorubicin (Sigma, St. Louis, MO) or exposure to 20 Grays of ionizing radiation. MEFs were prepared from Rb[f/f] mice (*Marino et al., 2000*) and cyclin D1[f/f]/D2[−/−]/D3 [f/f] mice (*Choi et al., 2012*). Rb and cyclin D inactivation was performed by addition of TAT-Cre protein (*Wadia et al., 2004*). U2OS-p16 cells (*Jiang et al., 1998*) were maintained in 1 μg/ml tetracycline to repress p16 expression. C2C12 myoblasts were differentiated into myotubes by incubating with DMEM plus 2% horse serum for 2 days.

### Generation of constructs and retroviruses

Human Rb[ΔCDK–HA] and murine Rb[ΔCDK–HA] were generated by changing all 15 Ser/Thr Cdk acceptor sites to Ala, with Ser567 and S561, respectively, left unaltered, with a HA tag placed on the N-terminus and C-terminus, respectively, and expressed from pCMV. Human Rb single Cdk sites were generated by individually adding back each single Cdk site to Rb[ΔCDK–HA]. Rb[2xCdk] retained T373, S811; Rb[3xCdk] retained T373, S612, S811; Rb[6xCdk] retained the N-terminal Cdk sites; Rb[9xCdk] retained spacer and C-terminal Cdk sites. Murine Rb[WT–HA] and Rb[ΔCdk–HA] MSCV retroviruses were generated from transfected HEK 293 cells and stored at −80°C. E2F constructs were expressed from pCMV and contained a C-terminal Myc tag. The Rb shRNA vector was generated by inserting a 3′ UTR region of the endogenous Rb mRNA (GCTTTGAACTGAAGACTAT) into pSM2c-scramble (*Stegmeier et al., 2005*).

## 2D IEF

2D-IEF was performed as described (*Ezhevsky et al., 2001*) by immunoprecipitating Rb and eluting in 7 M urea/2 M thiourea/2% CHAPS (pH 8.4), then loading onto the acidic end of a 3–10 immobiline strips (GE Healthcare) with the current ramped up from 200 V for 2 hr, 500 V for 1 hr, 800 V for 1 hr, 1000 V for 0.5 hr, 1200 V for 0.5 hr, 1400 V for 0.5 hr, 1600 V for 0.5 hr, 1800 V for 2.5 hr, and 2000 V for 2.5 hr. Second dimension was performed by soaking IEF strip in 2% SDS/6 M urea/75 mM Tris (pH 8.8), 29% (wt/vol) glycerol, and placing the strip on top of a 6% SDS-PAGE containing a single large well to accommodate the IEF strip with Mw marker side wells.

## Immunoprecipitations, immunoblotting and kinase assays

Co-immunoprecipitations were performed as described (*Ezhevsky et al., 1997*) using anti-Rb (G3-245, BD Pharmingen, San Jose, CA), anti-HA (3F10, Roche, Basel, Switzerland), anti-Myc (9E10, Developmental Studies Hybridoma Bank, Iowa City, IA), or anti-E1a (M73). Immunoblotting was performed as described (*Ezhevsky et al., 1997*) using anti-Rb (G3-245, BD Pharmingen), anti-HA (3F10, Roche), anti-actin (C4, Abcam), anti-E1a (13-S5; Santa Cruz), and anti-Myc (9E10, Developmental Studies Hybridoma Bank) antibodies. Rb immunoblots were performed using 6% SDS-PAGE for separation or 10% SDS-PAGE for quantification. All immunoblots were quantified utilizing ChemiDoc XRS (Bio-Rad, Hercules, CA) sub-saturating linear signals. Rb phospho-specific antibodies: T356-PO4 (AB4780, Abcam, Cambridge, England), S608-PO4 (2181, Cell Signaling, Danvers, MA), S612-PO4 (OPA1-03891, Thermo Scientific, Waltham, MA), S780-PO4 (3590, Cell Signaling), S807-PO4/S811-PO4 (9308, Cell Signaling), T821-PO4 (AB4787, Abcam), T826-PO4 (AB4779, Abcam), T821-PO4/T826-PO4 (sc-16669, Santa Cruz), T373 (AB52975, Abcam), S249-PO4/T252-PO4 (sc-16671, Santa Cruz). Immunoprecipitation-kinase assays were performed as described (*Ezhevsky et al., 1997*) using anti-CDK4 (C22), anti-CDK6 (C21), and anti-Cdk2 (M2) polyclonal antibodies (Santa Cruz).

## qRT-PCR and microarray analysis

qRT-PCR was performed as described (*Eguchi et al., 2009*) using 6-FAM labeled TaqMan probes (Dhfr, 00515663; Cdc6, 00488573; Ccna2, 01282245; p21, 00432448; β2M, 00437762; Mcm3, 00801867; Mcm5, 00484839; Life Technologies, Grand Island, NY). Mean values of triplicate samples were normalized to beta-2-microglobulin. Whole-genome microarray analysis was performed as described (*Eguchi et al., 2009*) using MouseWG-6 v2.0 BeadChips (Illumina, San Diego, CA) at Biogem core (UCSD). Heat maps were created with Cluster 3.0 and Java TreeView 1.1.3 and gene ontology classifications were based on DAVID Bioinformatics Resources (*Dennis et al., 2003*; *Huang da et al., 2009*). Full microarray data set has been submitted to GEO (GSE56453) http://www.ncbi.nlm.nih.gov/geo/query/acc.cgi?acc=GSE56453.

## Acknowledgements

We thank S Ezhevsky and J Adams for critical input; M Haas, P Hamel, S Elledge, J Baldassare, L Zhu, and the Developmental Studies Hybridoma Bank (U. Iowa) for constructs, antibodies and cell lines.

## Additional information

### Funding

| Funder | Grant reference number | Author |
|---|---|---|
| National Institutes of Health | CA169849 | Steven F Dowdy |
| Howard Hughes Medical Institute | | Steven F Dowdy |
| National Institutes of Health | CA108420 | Piotr Sicinski |
| Leukemia and Lymphoma Society | | Gary S Shapiro |
| National Institutes of Health | TG CA009523 | Anil M Narasimha |

The funders had no role in study design, data collection and interpretation, or the decision to submit the work for publication.

## Author contributions
AMN, Acquisition of data, Analysis and interpretation of data, Drafting or revising the article; MK, GSS, Conception and design, Acquisition of data, Analysis and interpretation of data, Drafting or revising the article; YJC, PS, Analysis and interpretation of data, Drafting or revising the article, Contributed unpublished essential data or reagents; SFD, Conception and design, Analysis and interpretation of data, Drafting or revising the article

## Additional files

### Major dataset

The following dataset was generated:

| Author(s) | Year | Dataset title | Dataset ID and/or URL | Database, license, and accessibility information |
| --- | --- | --- | --- | --- |
| Narasimha AM, Kaulich M, Shapiro GS, Choi YJ, Sicinski P, Dowdy SF | 2014 | Cyclin D:Cdk4/6 Activates Rb in Early G1 Phase by Mono-Phosphorylation | http://www.ncbi.nlm.nih.gov/geo/query/acc.cgi?acc=GSE56453 | Publicly available at NCBI Gene Expression Omnibus. |

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
