## [Decision Letter]

Thank you for sending your work entitled “Cyclin D:Cdk4/6 Activates Rb in Early G_1_ Phase by Mono-Phosphorylation” for consideration at *eLife.* Your article has been favorably evaluated by Tony Hunter (Senior editor), a Reviewing editor, and 3 reviewers.

The Reviewing editor and the other reviewers discussed their comments before we reached this decision, and the Reviewing editor has assembled the following comments to help you prepare a revised submission.

Narasimha et al. demonstrate that, contrary to dogma, cyclinD/CDK does not hypo- or hyper-phosphorylate pRb nor inactivate pRb. Their data suggest that pRb is activated in G1 phase by cyclinD/CDK dependent mono-phosphorylation at one of 14 CDK sites. Hyper-phosphorylation by cyclinE/CDK inactivates pRb to fully activate E2F-dependent transcription, which surprisingly is independent of cyclinD/CDK mono-phosphorylation. In addition their data indicates that DNA damage induced pRb activation requires cyclinD/CDK-dependent mono-phosphorylation. Finally they present data that suggests that un-phosphorylated pRb drives cell cycle exit (G0 entry) or prevents G1 entry. Based on these results they suggest a model in which an increase in cyclinD/CDK activity, observed in many cancers, 'primes' a cell for loss of cell cycle control by preventing cell cycle exit (or allowing G1 entry). This radically changes the way we think about the role of cyclinD/CDK in the regulation of pRb function.

Importantly, the data presented provide many new insights into the role of cyclinD/CDK in pRb regulation and challenge the current understanding of the role of cyclinD/CDK in proliferating, cell cycle exiting and DNA damaged cells, but also the positive feedback model in which cyclinD/CDK is the initial activator of E2F-dependent transcription resulting in accumulation of cyclinE/CDK. The experiments have been performed with care, but some of the conclusions drawn are not warranted by the data.

Given the extensive scope of the current study, the reviewers believe that the remaining issues can be adequately addressed by changes to the text of the manuscript without the need for further experimentation. Specifically:

1) The data shows that pRb is mono-phosphorylated by cyclinD in response to DNA damage, which is a major observation. However, they do not show if this is checkpoint dependent by, for example, using checkpoint protein kinase inhibitors (ATR/ATM, CHK1/CHK2). The inability to phosphorylate pRb in response to DNA damage results in activation of E2F-dependent transcription and genomic instability. They state that (line 419) the appearance of tetraploid cells indicates that the cells 'failed to activate the DNA damage checkpoint'. This is neither tested nor likely. If they want to conclude this, they should show that the checkpoint protein kinases are not activated in Rb-deleted or ΔCDK-Rb cells, which seems very unlikely. The reviewers recommend that the text of the manuscript should be revised to note these points.

2) The binding specificity of specific mono-phosphorylated pRb by certain E2Fs in Figure 2 is mentioned to be reproducible, but not shown. Can the authors elaborate on this point?

3) An alternative model for E2F-dependent transcriptional activation is not presented. The authors mention that 'at the restriction point cyclinE/CDK inactivates pRb'. Although it is beyond the scope of the paper to test alternative models for E2F-dependent transcriptional activation, the reviewers believe that this topic should be included in the discussion section of the manuscript.

---

## [Author Response]

*1) The data shows that pRb is mono-phosphorylated by cyclinD in response to DNA damage, which is a major observation. However, they do not show if this is checkpoint dependent by, for example, using checkpoint protein kinase inhibitors (ATR/ATM, CHK1/CHK2). The inability to phosphorylate pRb in response to DNA damage results in activation of E2F-dependent transcription and genomic instability. They state that (line 419) the appearance of tetraploid cells indicates that the cells 'failed to activate the DNA damage checkpoint'. This is neither tested nor likely. If they want to conclude this, they should show that the checkpoint protein kinases are not activated in Rb-deleted or ΔCDK-Rb cells, which seems very unlikely. The reviewers recommend that the text of the manuscript should be revised to note these points*.

We thank the reviewers for pointing out our poor wording in the original text and did not intend to suggest that Rb was upstream of ATR/ATM, etc. We completely concur that the DNA damage response is intact. Our investigation centered on the downstream consequences of cells either deleted for Rb or expressing a non-phosphorylatable ΔCdk Rb (un-phosphorylated Rb) that were exposed to doxorubicin or ionizing radiation and that failed to adequately arrest resulting in high numbers of tetraploid cells. This experiment showed that mono-phosphorylated Rb, generated by cyclin D:Cdk4/6, was functional and required to prevent inappropriate cell cycle progression and E2F transcription *downstream* of an ATR/ATM DNA damage response. We have clarified the text accordingly.

*2) The binding specificity of specific mono-phosphorylated pRb by certain E2Fs in*
Figure 2
*is mentioned to be reproducible, but not shown*. *Can the authors elaborate on this point?*

This specific comment was meant to reinforce that the rather subtle differences observed between E2F2 and E2F3 binding to specific mono-phosphorylated Rb isoforms was consistently observed in three independent experiments, whereas E2F1 and E2F4 showed much larger binding differences.

*3) An alternative model for E2F-dependent transcriptional activation is not presented. The authors mention that 'at the restriction point cyclinE/CDK inactivates pRb'. Although it is beyond the scope of the paper to test alternative models for E2F-dependent transcriptional activation, the reviewers believe that this topic should be included in the discussion section of the manuscript*.

While our study determined the role of cyclin D:Cdk4/6 in mono-phosphorylating Rb, it leaves wide open the question of what the rate-limiting switch-like mechanism is to activate cyclin E:Cdk2, the first domino in Rb inactivation. Cyclin D:Cdk4/6 activity combined with other signal transduction pathway mutations contributes to increased cellular metabolism that we speculate is monitored by an unknown metabolic sensor. Once the metabolic threshold has been exceeded, the sensor activates cyclin E:Cdk2 resulting in Rb inactivation by hyper-phosphorylation, induction of E2F target gene transcription and progression across the Restriction Point into late G_1_ phase. We have expanded the Discussion section to include our thoughts on cyclin E:Cdk2 activation.

In conclusion, we maintain that cyclin D:Cdk4/6 complexes strictly mono-phosphorylate Rb during early G_1_ phase to generate 14 independent isoforms that bind E2F1-4 and are functionally active during a DNA damage response. We also assert that there is no progressive multi-phosphorylating, hypo-phosphorylation of Rb. We believe that both the depth of analyses and the magnitude of impact on understanding G_1_ cell cycle progression warrants publication in *eLife*.